# Obscuring Data Contamination Through Translation: Evidence from Arabic Corpora

## Abstract

Data contamination threatens the validity of Large Language Model (LLM) evaluation by allowing models to exploit memorized benchmark content rather than demonstrating true generalization. While existing detection methods focus primarily on English datasets, little is known about how contamination manifests in multilingual contexts. In this paper, we investigate contamination dynamics by fine-tuning several open-weight LLMs on varying proportions of different Arabic datasets and evaluating them on the original English benchmarks. To probe memorization, we extend the Tested Slot (TS)-Guessing method with a choice-reordering strategy, enabling us to disentangle genuine reasoning from contaminated recall. Our results show that while translation into Arabic conceals traditional contamination signals, models still benefit from exposure to contaminated data, particularly those with stronger Arabic capabilities. This demonstrates that translations can mask but not eliminate contamination, creating a dangerous blind spot in current evaluation practices. To address this, we propose a Translation-Aware Contamination Detection framework, which checks contamination across multiple translated versions of benchmarks rather than relying on English alone. Together, our findings highlight the need for multilingual contamination-aware pipelines to ensure fair, transparent, and reproducible evaluation of LLMs.

## 1 Introduction

Large Language Models have achieved remarkable success across a wide range of natural language understanding and generation tasks. Their progress has been largely measured using standardized benchmarks such as MMLU, HellaSwag, ARC, and others, which aim to test reasoning and knowledge beyond mere memorization. However, recent research has raised serious concerns about data contamination, a phenomenon where benchmark data—or closely related material—appears in the training corpus of the evaluated model. Contamination undermines the validity of benchmark-based evaluation, as models may exploit memorized content rather than demonstrating genuine generalization ability.

While several detection methods have been proposed, ranging from corpus-level search tools to guided prompting strategies, they remain fragmented, computationally expensive, and limited in scope. Moreover, transparency is hindered by the lack of disclosure of training data from major proprietary models, making it nearly impossible for the community to establish consensus on the reliability of evaluations. Beyond this technical challenge, contamination has broader implications for reproducibility, fairness, and the trustworthiness of machine learning as a scientific discipline.

In this work, we investigate contamination from a multilingual perspective. Specifically, we ask whether translating benchmarks into a low resources language—in our case, Arabic—can act as a natural barrier to contamination or whether translation merely conceals memorization effects. We fine-tune open-weight models on different benchmarks with varying proportions of their Arabic-translated test set and evaluate their performance on the original English benchmark. To disentangle genuine reasoning from memorization, we employ the TS-Guessing method, extended with a multiple-choice reordering trick. Our analysis reveals that while translation obscures traditional contamination signals, models still benefit disproportionately from exposure to contaminated data, particularly those with stronger Arabic capabilities. This highlights a critical blind spot in current

contamination detection approaches and calls for contamination-aware evaluation pipelines that explicitly account for multilingual dynamics.

## 2 LITERATURE REVIEW

In the context of LLMs, data contamination refers to any instance where a LLM model preprocessed an evaluation benchmark during its training. Contamination allows models to cheat by memorizing a dataset rather than displaying true generalization. Beyond being a technical concern, contamination threatens the reliability of evaluation pipelines, raising questions of reproducibility, fairness, and transparency in the broader trustworthy ML landscape.

### 2.1 FORMS OF CONTAMINATION

Contamination can occur in many forms, and each has its own implications. In this section we explain the different types of contamination and their implications on LLM evaluations.

#### 2.1.1 GUIDELINE CONTAMINATION

Guideline contamination is when the guidelines to a specific dataset are seen by a model during training. These guidelines are usually publicly found on the internet even for datasets that are not available for public use. Detailed guidelines can even include samples from the dataset. Models exposed to these guidelines during training have an advantage over those without such information. Such exposure invalidates zero-shot and few-shot evaluations. Sainz et al. (2023)

#### 2.1.2 RAW TEXT CONTAMINATION

Raw text contamination takes place when the pre-annotated raw text is seen by the model. Typical examples include datasets that are based on Wikipedia texts. Models already exposed to Wikipedia during training have an advantage over other models on tasks such as name entity recognition. Moreover, datasets such as CNN/DailyMail are compromised since they are scraped from publicly available news websites. Sainz et al. (2023)

#### 2.1.3 ANNOTATION CONTAMINATION

Annotation contamination happens if the labels of a benchmark are exposed during training. When the evaluation sets are compromised, the entire experiment is invalidated, and when the training data is seen during training, claims of zero-shot or few-shot performance are worthless. Sainz et al. (2023)

Beyond these forms, recent work has argued that contamination can also indirectly amplify fairness and bias concerns. For example, if benchmark data associated with specific dialects, demographics, or regions is memorized, models may artificially appear more capable in those subdomains while failing to generalize elsewhere. This links contamination to bias amplification and evaluation consistency problems observed in safety-aligned models Fraser et al. (2025).

### 2.2 CHALLENGES

#### 2.2.1 DATA CONTAMINATION IS NOT PROPERLY REPORTED

Although contamination analysis has become an important part of evaluating LLMs, it is usually conducted internally by the LLM developers, and it lacks transparency and completeness Li (2023). For example, OpenAI contamination studies only focus on the training data and omit data used in fine-tuning stages Li (2023); OpenAI (2023). Furthermore, LLAMA-2 only reported contamination analysis results for 2 of the 20+ benchmarks used in their evaluation. This lack of transparent reporting undermines reproducibility, since external researchers cannot independently verify claims, creating a gap between industry and academic evaluation practices.

### 2.2.2 Contamination Analysis Is Not Conducted Correctly

Companies are committing mistakes that invalidate their contamination analysis. For example, OpenAI OpenAI (2023) conducted their contamination analysis post pretraining the GPT-4 model. Although their evaluation scores are calculated after the contaminated examples were removed from the benchmarks, having the model exposed to those examples during training invalidates all claims of zero-shot and few-shot results.

### 2.2.3 Proprietary Models Do Not Disclose Their Training Data

OpenAI and Mistral OpenAI (2023); Jiang et al. for example, are not disclosing the training data used for their models (GPT-4 and Mixtral). This makes it more difficult to answer the question "Was this seen during training" Marone & Van Durme (2023).

### 2.2.4 Pre-Training Datasets Are Large

The datasets used to train the models can be in the order of terra bytes in size, which poses a challenge for efficiently indexing and querying these datasets for benchmark data contamination Li (2023); Golchin & Surdeanu (2023b).

## 2.3 Contamination Detection Methods

Many attempts for determining data contamination rely on researchers building search tools for popular training corpuses and using these tools to make queries for exact or fuzzy matches with samples from evaluation benchmarks.

Piktus et al. built a 2.78 terra byte search tool for the Roots corpus used to train all Bloom models. The tool was built using the BM25 index algorithm and was successfully used to determine that Bloom should not be evaluated on the XNLI dataset, and that the Roots corpus contains more languages than intended by the authors. Perhaps a similar tool can be used to test other datasets for contamination.

Li (2023) base their data contamination studies on locating benchmark samples in the Common Crawl corpus because this corpus comprises most of the pretraining data for most LLM models. They first used the Bing Search API to check if the benchmark examples are located online because samples that are online are likely to be included in the Common Crawl. If the Bing Search API returned a positive result, they verified if the sample is in the Common Crawl Corpus by searching the corpus for the URL of the page that includes the benchmark example. They used the METEOR score to evaluate similarity between the benchmark examples and the search results and considered examples with scores above 0.75 to be contaminated. They investigated the Common Crawl time windows used to train popular models including LLAMA, LLAMA-2, Mistral, Yi, Qwen, and Baichuan2 and showed that the data used to train these models is contaminated with examples from the MMLU, Hellaswag, ARC, and C-Eval benchmarks. Furthermore, their findings indicate that the contamination levels grow through time as more and more benchmark data is accessible on the internet and included as part of Common Crawl. Finally, they noted that larger models can better exploit data contamination than smaller models leading to more significant score boosts. While the authors did not test GPT models and OpenAI does not explicitly disclose the training data for GPT-3.5 and GPT-4, GPT-3 used Common Crawl for more than 80% of its training data Li (2023). Therefore, similar contamination is highly plausible for GPT models.

Marone & Van Durme (2023) constructed a Bloom filter based membership testing tool for The Pile corpus (the text subset of the Common Crawl corpus). Bloom filters are like Hash tables except that they only store the Hash without the original text reducing the memory and computation time needed to index and query. Their data structure requires only 27 GB of disk storage despite The Pile dataset being 825 GB. They tested their tool on the XSum, TIFU, and AMR2.0 datasets and where able to find a lot of exact matches in The Pile corpus indicating contamination.

While the above methods were successful in locating instances of contamination in various training corpuses, they are still computationally expensive and cannot be used to identify instances of contamination when companies do not properly disclose their training data. Therefore, methods were proposed to identify contamination by only querying the trained models.

Table 1: The Min-K% Prob method applied to some benchmarks and models

| Model | ARC | HellaSwag | MMLU | TruthfulQA | Winogrande | GSM8K |
|---|---|---|---|---|---|---|
| Mistral-7B-v0.1 | 0.54 | 0.51 | 0.46 | 0.75 | 0 | 0.91 |
| Mistral-7B-Instruct-v0.2 | 0.06 | 0.21 | 0.17 | 0.48 | 0 | 0.95 |
| Zephyr-7B-Beta | 0.06 | 0.15 | 0.18 | 0.37 | 0 | 0.82 |
| Falcon-7B-Instruct | 0.11 | 0.17 | 0.22 | 0.46 | 0 | 0.79 |
| Yi-6B | 0.28 | 0.32 | 0.30 | 0.62 | 0.02 | 0.94 |

Golchin & Surdeanu (2023b) used guided instructions to force the LLMs to produce exact match or near exact match samples from different NLP benchmarks. Their guided instructions included the name of the dataset of interest, the partition (train, validation, test, etc), a portion of the benchmark sample of interest, and the label. Then, the LLMs generated a possible continuation of the benchmark sample, and the generated samples were compared to the benchmark sample. High matching between the generated and benchmark samples indicates that the LLM was exposed to the benchmark during its training. Their results showed that ChatGPT training data is contaminated with the AG News, WNLI, IMDB, RTE, and XSum datasets. The results are not exhaustive since the authors did not investigate other datasets.

Golchin & Surdeanu (2023a) improved the guided instructions approach of Golchin & Surdeanu (2023b) to factor in the probabilistic nature of LLMs, which prevents them from always generating exact or near exact matches. They enhanced the guided prompt by giving the LLM multiple choices and asking it to pick the choice that corresponds to an exact match instance in the benchmark dataset. One of the choices would always be the unaltered benchmark sample and the other choices would be paraphrases. Their results align with Golchin & Surdeanu (2023b) and indicate that GPT models are contaminated with AG News, WNLI, IMDB, RTE, SAMSum and XSum datasets.

Shi et al. (2023) determine contamination by calculating a metric they coin as the Min-K% Prob. The Min-K% Prob first uses the LLM to get the probabilities for each token in a text X, and then it selects the K% tokens with minimum probabilities and calculates their average log likelihood. The text is likely in the training data if the average likelihood is high. Their results show the following: (1) Contamination detection becomes easier as model size and text length increases. (2) GPT-3 was trained on copyrighted books form the Books3 corpus.

Table 1 shows the results of Min-K% Prob applied to the Huggingface benchmarks on some models. It is evident that the models demonstrate contamination especially on the GSM8K benchmark where most models are more than 90% likely to be contaminated.

Although these methods vary in computational cost and coverage, none currently provide a scalable, standardized framework for the community. There remains a need to move from ad-hoc tools toward reproducible, benchmark-independent contamination detection pipelines that can be shared, audited, and extended across different research groups.

### 2.4 What Can Be Done Regarding Contamination

Combating data contamination is still in its infancy, and Sainz et al. (2023) recommend focusing efforts on on the following.

- Developing automatic detection measures.

- Building a registry for offenders with the evidence of contamination.

- Encouraging authors to use tools that ensure avoiding contamination as possible.

- Highlighting data contamination during peer review, and flagging published work with evidence of contamination.

The suggestions above all rely on having and developing reliable detection measures, and none of the methods in section 2.3 are proven to provide 100% detection.

Furthermore, Li (2023) have shown that the following mitigation measures suggested by Jacovi et al. (2023) are not very effective.

- **Blocklist benchmark sources during training data scraping:** the authors argue that while avoiding benchmark websites while collecting data reduces the chances of contamination, even the most aggressive blocklisting approaches result in only avoid 21% of contamination cases. The authors explain that blocklisting is ineffective since benchmark data rapidly spreads over the internet making it necessary to keep on updating the blocklists regularly.

- **Avoid input-output contamination:** the authors' results show that that not having the answers to benchmark questions can prevent exact memorization, and the models' performance is worse on input-only contamination compared to on input-output contamination. However, their results also show that input-only contaminated training sets provide an unfair advantage relative to clean training sets. Furthermore, the authors do not explain how input-output contamination can be avoided in the first place especially that most benchmarks are being published with the inputs and outputs included in the same data structure.

- **Protect the benchmark test data and do not distribute it:** According to the authors' results, hiding the test data can reduce the effects of data contamination, but this makes third party evaluation platforms resort to using the publicly available validation set to access models' performance.

On the level of current leader-boards, little is being done to filter our model evaluations that may be affected by contamination. The Huggingface Open LLM leader-board allows for flagging models models that show exceedingly good results without reasonable justification. While this prevents any intentional or accidental training on the complete evaluation sets to artificially boost results, slight manipulations that could slightly nudge a model slightly on the leader-board could go undetected. Moreover, The BigCode leader-board indicates what registered scores have been obtained from external evaluations, but such indications do not give any information about the validity of the scores.

Therefore, since the above methods are not fully effective, methods in section 2.3 are not proven to be 100 % reliable, and larger pretraining datasets are difficult to query for contamination Li (2023); Golchin & Surdeanu (2023b), we conclude that contamination can be mitigated but not fully prevented. However, it remains the responsibility of the authors of scraped datasets to ensure that the data they produce is decontaminated as much as possible using state of the art techniques prior to publishing it. Also, the continuous efforts to decontaminate and publish clean versions of pretraining datasets provide LLM developers readily available clean data. Such cleaning efforts impose pressure on LLM developers to stop using raw contaminated data since it is more difficult to justify using such data when clean data is available. Furthermore, it is the responsibility of LLM developers to report what clean datasets they used in training the model, or to disclose data cleaning methods they used in preparing their proprietary datasets.

Taken together, the literature suggests that while detection and mitigation tools are advancing, current approaches remain fragmented, non-standardized, and often computationally prohibitive. This creates a gap for methods that can both scale with modern pretraining corpora and provide transparent, reproducible contamination analysis. Moreover, prior studies have primarily focused on contamination in English benchmarks, overlooking how translations into other languages may alter contamination dynamics. Our work addresses this gap by investigating whether translating benchmarks into Arabic can act as a natural barrier against contamination, and by combining this with targeted detection methods such as TS-Guessing to disentangle genuine reasoning ability from memorization effects.

## 3 METHODOLOGY

### 3.1 MODELS AND TRAINING SETUP

We fine-tuned four open-weight instruction models: `Llama-3.2-1B-Instruct`, `Mistral-7B-Instruct-v0.2`, `Gemma-3-1B-it`, and `Qwen3-1.7B`. For each dataset $d \in \{\text{MMLU}, \text{XQuAD}, \text{MLQA}\}$ we created four training conditions:

$$\mathcal{D}^d_{\text{train}}(p) = \mathcal{D}^d_{\text{EN}} \cup \mathcal{D}^d_{\text{AR}}(p), \qquad p \in \{0, 10\%, 50\%, 100\%\},$$

where $\mathcal{D}^d_{\text{EN}}$ is the English split (MMLU: English test items formatted as MCQ; XQuAD/MLQA: English QA), and $\mathcal{D}^d_{\text{AR}}(p)$ is a proportion-$p$ subset of the Arabic counterpart (MMLU: Arabic trans-

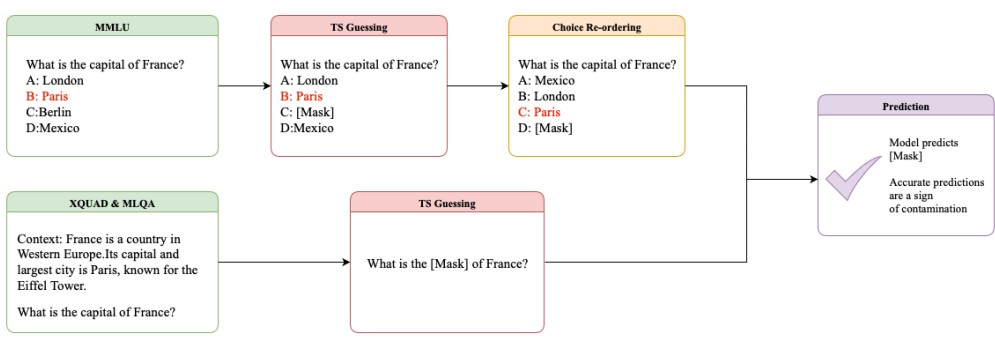

Figure 1: TS-Guessing across datasets. Top: MMLU (MCQ) with choice re-ordering then masking a choice; index-letter recall after shuffling is a contamination cue. Bottom: XQuAD/MLQA (extractive QA) with a masked token in the question; exact recovery suggests memorization.

lations of the test items; XQuAD/MLQA: Arabic split). Thus for each $d$ we train four separate models: EN-only ($p=0$), EN+AR10, EN+AR50, and EN+AR100.

All runs use parameter-efficient fine-tuning (LoRA/PEFT) with identical optimizer, schedule, context length, and batch policy across models and conditions to ensure fair comparison.

For extended dataset statistics and overlap/length analyses, consult Appendix B.

## 3.2 EVALUATION PROTOCOL

We evaluated each fine-tuned model on the standard evaluation split(s) for its dataset using the Evaluation Harness Gao et al. (2021), and reported these metrics for each dataset:

- **MMLU (English).** Macro-averaged accuracy over subjects using the Evaluation Harness Gao et al. (2021).

- **XQuAD / MLQA.** ROUGE-L F1 on the English splits.

- **Contamination sensitivity.** Performance trend as the contamination bucket increases (10% $\rightarrow$ 100%) to assess dependence on exposed evaluation content.

## 3.3 CONTAMINATION PROBES VIA TS-GUESSING

We probe memorization with TS-Guessing (Deng et al., 2024) under the same contamination conditions per dataset $p \in \{10, 50, 100\}\%$ and for $d \in \{\text{MMLU}, \text{XQuAD}, \text{MLQA}\}$ as shown in figure 1.

**MMLU (MCQ):** For each item, we (i) randomly shuffle choices (A,B,C,D), then (ii) mask the *text* of one incorrect answer and prompt the model to fill the mask. If a model reproduces the pre-shuffle letter/index (e.g., outputs "D" when the correct answer moved to position A) or the text of the masked choice, this indicates reliance on memorized index patterns rather than reasoning about content.

**XQuAD/MLQA (extractive QA):** We mask a critical token in the question (e.g., "What is the [MASK] of France?") while keeping the context unchanged, and prompt the model to predict the masked token. Consistently exact recovery of the masked token signals possible leakage.

## 3.4 ANALYSIS & METRICS

we compute:

- **Exact Match (EM)- XQUAD and MLQA only.** $\text{EM} = \frac{1}{N} \sum_i \mathbf{1}\{\hat{y}_i = y_i\}$, where $y_i$ is the masked text (XQuAD/MLQA masked token).

Table 2: Results of English MMLU, XQuAD, and MLQA using the Evaluation Harness.

| Contam. (%) | Mistral-7B-Instruct-v0.2 | | | Gemma-3-1B-it | | | LLaMA-3.2-1B-Instruct | | | Qwen3-1.7B | | |
| --- | --- | --- | --- | --- | --- | --- | --- | --- | --- | --- | --- | --- |
| | MMLU | XQuAD | MLQA | MMLU | XQuAD | MLQA | MMLU | XQuAD | MLQA | MMLU | XQuAD | MLQA |
| 0 | 0.577 | 0.302 | 0.444 | 0.220 | 0.364 | 0.474 | 0.332 | 0.364 | 0.443 | 0.553 | 0.457 | 0.162 |
| 10 | 0.580 | 0.455 | 0.446 | 0.244 | 0.481 | 0.494 | 0.381 | 0.459 | 0.520 | 0.560 | 0.429 | 0.409 |
| 50 | 0.600 | 0.272 | 0.373 | 0.261 | 0.577 | 0.411 | 0.389 | 0.558 | 0.437 | 0.562 | 0.510 | 0.157 |
| 100 | 0.690 | 0.114 | 0.379 | 0.284 | 0.606 | 0.471 | 0.431 | 0.569 | 0.456 | 0.581 | 0.564 | 0.153 |

- **ROUGE-L F1.** Partial-overlap score between $\hat{y}_i$ and $y_i$, reported for XQuAD/MLQA when EM is not met.
- **Index-recall rate (IDR) - MMLU only.** $\mathrm{IdxRec} = \frac{1}{N}\sum_i \mathbf{1}\{\hat{\ell}_i = \ell_i^{\mathrm{pre\text{-}shuffle}}\}$, indicating the fraction of predictions that echo the original answer letter after re-ordering (a strong contamination signal).

## 4 RESULTS

### 4.1 EVALUATION

**Overall trends.** Across all models, *MMLU* exhibits a generally monotonic increase as contamination rises from $0\% \rightarrow 100\%$ (e.g., Mistral: $0.577 \rightarrow 0.690$, Gemma: $0.220 \rightarrow 0.284$, LLaMA: $0.332 \rightarrow 0.431$, Qwen: $0.553 \rightarrow 0.581$). This pattern is consistent with contamination-driven memorization: when benchmark content or near-duplicates appear during training, closed-book multiple-choice accuracy tends to inflate without necessarily reflecting improved reasoning.

**Span extraction vs. memorization.** For *XQuAD* and *MLQA* (extractive QA), trends are more model-specific and often non-monotonic:

- **Gemma & LLaMA.** Both show steady XQuAD gains with contamination (Gemma: $0.364 \rightarrow 0.606$, LLaMA: $0.364 \rightarrow 0.569$). However, *MLQA* is *non-monotonic* (Gemma: 0.474, 0.4936, 0.4109, 0.4707; LLaMA: 0.4430, 0.5198, 0.4371, 0.4563), suggesting that early leakage (e.g., $10\%$) can boost cross-lingual span fidelity, but further contamination may overfit to distributional quirks that do not transfer as well to MLQA.
- **Mistral.** MMLU rises strongly, yet XQuAD *collapses* beyond $10\%$ ($0.455 \rightarrow 0.272 \rightarrow 0.114$), and MLQA dips ($0.4460 \rightarrow 0.3727 \rightarrow 0.3789$). This divergence (closed-book up, extractive down) is consistent with memorization that helps option selection while harming calibration and span localization under distribution shift.
- **Qwen.** XQuAD dips at $10\%$ then recovers ($0.457 \rightarrow 0.4291 \rightarrow 0.510 \rightarrow 0.564$), whereas MLQA spikes at $10\%$ ($0.1621 \rightarrow 0.4086$) and then *collapses* ($0.1565$, $0.1534$). The anti-correlation between XQuAD and MLQA at moderate/high contamination hints at dataset-specific leakage or language/domain mismatch: what is memorized may help one English QA set while degrading generalization to MLQA's cross-lingual paraphrases and contexts.

**Non-monotonic MLQA suggests fragile transfer.** The frequent "peak-at-$10\%$" in MLQA (Gemma, LLaMA, Qwen) followed by a decline at $50\%$ and partial recovery or further decline at $100\%$ indicates that small amounts of overlap can aid cross-lingual extraction (easier anchor points), while heavier contamination may overfit to some unique lexical or formatting attributes that do not carry over to MLQA. In other words, contamination can improve surface-form familiarity without improving cross-lingual grounding.

### 4.2 CONTAMINATION ANALYSIS

Across contamination levels $p \in \{10, 50, 100\}\%$, the models exhibit approximately equal performance on all evaluated benchmarks. This near-flat trend indicates that Arabic→English translation is effectively masking contamination effects rather than removing them. In typical same-language settings, increasing $p$ would be expected to induce noticeable shifts (e.g., steady gains or degradations); here, however, translation perturbs surface forms while preserving sufficient semantic signal

Table 3: TS-Guessing results on MMLU (MCQ) and XQuAD (QA) at different contamination levels.

(a) Overall TS-Guessing (MCQ) on MMLU

| Model | 10% | | 50% | | 100% | |
|---|---|---|---|---|---|---|
| | IDR | RL-F1 | IDR | RL-F1 | IDR | RL-F1 |
| LLaMA-3.2-1B-Instruct | 0.287 | 0.017 | 0.643 | 0.006 | 0.410 | 0.006 |
| Qwen3-1.7B | 0.261 | 0.001 | 0.251 | 0.019 | 0.208 | 0.014 |
| Gemma-3-1B-it | 0.350 | 0.054 | 0.029 | 0.115 | 0.005 | 0.059 |
| Mistral-7B-Instruct-v0.2 | 0.000 | 0.039 | 0.000 | 0.043 | 0.001 | 0.037 |

(b) TS-Guessing (QA) on XQuAD

| Model | 10% | | 50% | | 100% | |
|---|---|---|---|---|---|---|
| | EM | RL-F1 | EM | RL-F1 | EM | RL-F1 |
| LLaMA-3.2-1B-Instruct | 0.001 | 0.002 | 0.005 | 0.005 | 0.008 | 0.010 |
| Qwen3-1.7B | 0.000 | 0.000 | 0.000 | 0.000 | 0.003 | 0.003 |
| Gemma-3-1B-it | 0.017 | 0.018 | 0.013 | 0.014 | 0.005 | 0.007 |
| Mistral-7B-Instruct-v0.2 | 0.103 | 0.113 | 0.093 | 0.101 | 0.074 | 0.083 |

for task success, compressing the observable differences across $p$. The consolidated results in Tables 2 and 3a show that scores remain broadly stable as $p$ increases, supporting the interpretation that translation conceals leakage that would otherwise be apparent.

This stability should not be taken as evidence of decontamination. Rather, it suggests that the evaluation pipeline, once mediated by translation, underestimates the extent to which exposure influences downstream behavior. Because translation alters morphology, word order, and token boundaries while keeping core semantics largely intact, performance can remain steady even when exposure changes substantially. In practice, the flatness across $p$ seen in Tables 2 and 3a is therefore best understood as a limitation of surface-form detectability under translation, not as confirmation of contamination-free generalization.

### 4.3 DISCUSSION

Across models, MMLU increases monotonically with contamination, whereas XQuAD/MLQA fluctuate non-monotonically. This divergence indicates contamination-driven memorization of surface options rather than stronger span grounding or cross-lingual reasoning. The dataset statistics (low context–question lexical overlap; short yet non-trivial answer spans) further suggest that extractive QA hinges on fine-grained semantic alignment that is not uniformly improved by exposure.

The embedding figure shows that Arabic→English translations remain close to their English originals in representation space, with high cosine similarity

$$s = \cos\left(\mathbf{e}^{ar \rightarrow en}, \mathbf{e}^{en}\right).$$

Thus, translation perturbs tokens and word order while preserving meaning, compressing observed performance differences across $p$. In other words, translation $\neq$ decontamination: leakage persists through semantics, inflating closed-book accuracy while leaving QA transfer fragile.

Future audits should quantify representation overlap (e.g., cosine/CCA by layer), use provenance-controlled test sets, and pair accuracy with memorization probes (e.g., TS-Guessing) to disentangle true generalization from contamination-induced familiarity.

## 5 TRANSLATION-AWARE CONTAMINATION DETECTION

Our results highlight a major limitation in current contamination detection methods: when benchmarks are translated into another language, models can still exploit memorized content, yet standard English-only checks fail to capture this. To address this gap, we outline a Translation-Aware Contamination Detection (TACD) framework, designed to identify contamination signals across multilingual benchmark variants.

### 5.1 CONCEPT

TACD builds on the observation that contamination is not strictly tied to surface forms, but rather to semantic patterns that persist across languages. If a model has memorized benchmark content, contamination signals—such as reproducing answer strings or recalling original choice indices—should reappear even after translation and paraphrasing.

## 5.2 Key Components

- **Cross-Translation Benchmarking:** Construct multiple benchmark variants through both human and machine translation (e.g., Arabic, French, Chinese), preserving semantic meaning while introducing surface-level diversity. This ensures that contamination detection is not limited to a single language representation.

- **TS-Guessing Across Variants:** Extend the TS-Guessing method with and without the reordered answer choices to all translated benchmarks. If a model reproduces memorized answers consistently across different translations, this signals contamination rather than true reasoning ability.

- **Back-Translation Consistency:** Translate model outputs from other languages back into English and compare them with the original benchmark responses. Consistent alignment across languages suggests the model is recalling stored knowledge rather than reasoning through the question.

## 5.3 Discussion and Implications

While TACD is promising, it demands extensive multilingual resources, high-quality translations, and substantial compute. Building and maintaining multi-language benchmark variants is non-trivial: it requires linguistic expertise for semantic fidelity and large-scale resources for consistent evaluation. Translation also introduces noise—shifts in phrasing, cultural references, or syntax—that can blur whether performance differences reflect genuine reasoning or artifacts. Checking consistency across multiple translations adds complexity, since variability can arise from both the model and the translation pipeline, making it hard to separate memorization from cross-lingual generalization.

Accordingly, we offer TACD as a forward-looking blueprint rather than a complete implementation; deploying it at scale would require shared infrastructure and community resources.

Even so, TACD underscores the need to move beyond English-centric checks. Translation alone is not decontamination—it can mask contamination while preserving memorization. A translation-aware pipeline would better guard against hidden effects, curb overestimation of model ability, and support trustworthy multilingual benchmarks, advancing fair, transparent, and resilient LLM evaluation.

## 6 Conclusion

Data contamination remains a pervasive and unresolved challenge in the evaluation of LLMs. Our study demonstrates that while models exhibit measurable performance gains when fine-tuned on translated benchmark data, these improvements often mask underlying contamination rather than reflecting true generalization. Arabic translations, in particular, create an illusion of decontamination: they obscure exact string matches and evade standard detection tools, yet contaminated knowledge continues to influence model behavior. This finding reveals an overlooked risk in multilingual evaluation settings, where contamination may persist in subtler and harder-to-detect forms.

We conclude that combating contamination requires moving beyond English-centric methods. Future research should focus on building standardized, reproducible pipelines that can operate across languages, model families, and pretraining corpora. Furthermore, community-driven efforts to produce and maintain clean pretraining datasets, alongside transparent reporting from model developers, are essential for restoring trust in benchmark-based evaluation. By highlighting the interplay between translation and contamination, our work underscores the need to rethink evaluation practices in order to ensure fairness, reproducibility, and scientific integrity in the era of large-scale language models.

## 7 Reproducibility Statement

All hyperparameters and training/evaluation settings required to reproduce our results are enumerated in Appendix A. Upon acceptance, we will release the full codebase and scripts necessary to replicate all experiments and tables.

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

## A  HYPERPARAMETERS

Table 4: Common fine-tuning hyperparameters (HuggingFace `TrainingArguments`).

| Argument | Value |
|---|---|
| per_device_train_batch_size | 16 |
| per_device_eval_batch_size | 16 |
| gradient_accumulation_steps | 2 |
| optim | adamw_torch |
| num_train_epochs | 5 |
| evaluation_strategy | no |
| eval_steps | 0.2 |
| logging_steps | 1 |
| warmup_steps | 10 |
| logging_strategy | steps |
| learning_rate | 2e-4 |
| fp16 | True |
| report_to | none |

Table 5: TS-Guessing hyperparameters consolidated across datasets. A dash (—) indicates not applicable or not used.

| Setting | MMLU | XQuAD/MLQA |
|---|---|---|
| SPLIT | test | validation |
| SUBJECTS | [] | — |
| SEED | 1337 | — |
| MAX_ITEMS / MAX_EXAMPLES | None | None |
| MASK_TOKEN | — | [MASK] |
| MAX_NEW_TOK(ENS) | 32 | 8 |
| TEMPERATURE | 0.0 | 0.0 |
| TOP_P | 1.0 | None |
| TOP_K | — | None |
| *Filtering thresholds (MMLU only)* | | |
| ROUGE_SIM_THRESHOLD | 0.65 | — |
| DROP_BOOLEAN_QS | True | — |
| DROP_MATHY_QS | True | — |

# B DATASET COMPLEXITY AND DESCRIPTIVE STATISTICS

We first report basic corpus characteristics to contextualize later analyses. **MMLU** is substantially larger than **XQUAD** and **MLQA** and is the only dataset with explicit subject labels, whereas the QA sets provide extractive question–answer pairs without subjects. Across the QA corpora, context–question lexical overlap is modest (means ≈6–8% with upper-tail percentiles below 25%), indicating that questions are not mere rephrasings of their contexts. Answer spans in **MLQA** are short but non-trivial (median ≈2 words; upper tail up to ∼19), while **XQUAD** features very concise spans with low percentiles, aligning with its design toward succinct extractive answers. These descriptive trends, together with vocabulary size and type–token ratios, frame the comparative difficulty and stylistic differences highlighted in the figures below.

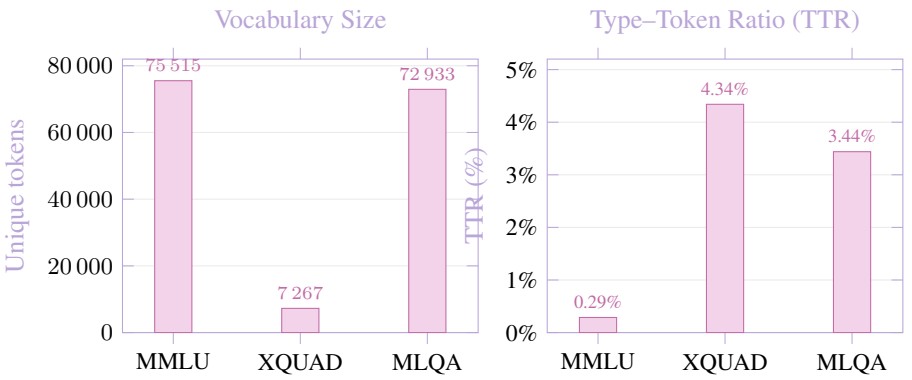

Figure 2: Dataset statistics: vocabulary size and TTR for **MMLU**, **XQUAD**, and **MLQA**.

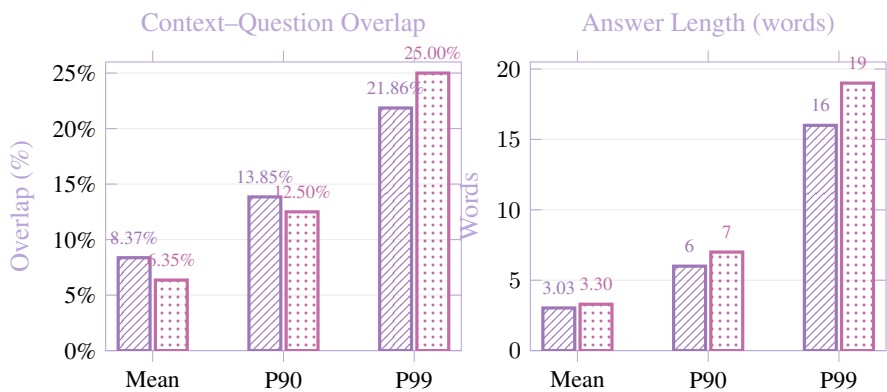

Figure 3: Question–context overlap and answer length statistics. Hatched plum bars denote **XQUAD (EN)**; dotted dusty-rose bars denote **MLQA**.

Figure 4 visualizes how closely Arabic→English translations align with the original English prompts at the *representation* level across subjects (e.g., *high_school_mathematics*, *professional_medicine*). Each band carries items from a similarity bin on the left to a subject on the right; thicker bands indicate more items. The concentration of flow from upper similarity bins toward many subjects suggests that translation preserves semantics strongly (high cosine similarity between embeddings of translated vs. original items). Where bands spread across lower bins, the subject shows more variability—indicating weaker preservation for a subset of items. Overall, the figure supports our claim that translation maintains meaning in embedding space, which can mask contamination signals that rely on surface-form differences.

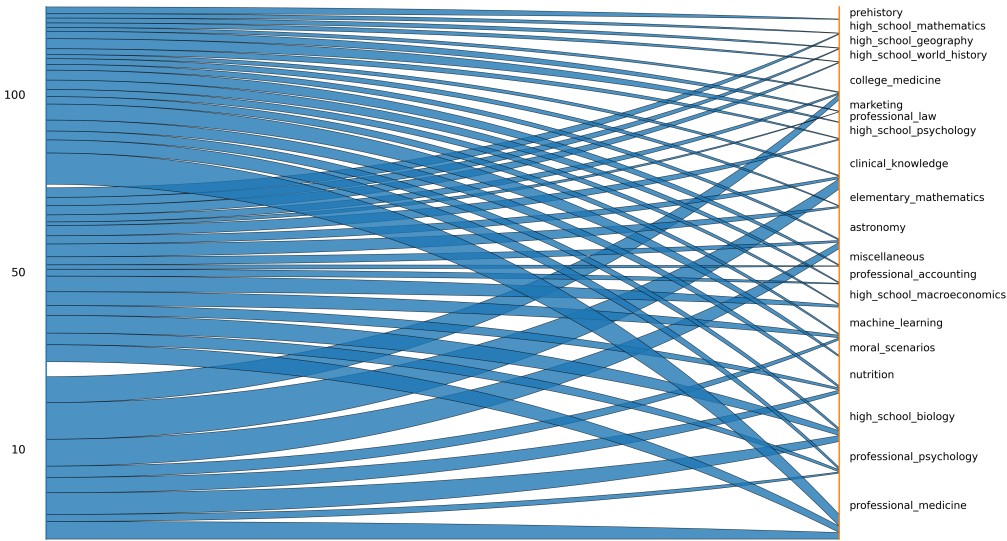

Figure 4: Flow diagram mapping Arabic→English translated items (left) to MMLU subject labels (right). Band thickness encodes the number of items; higher positions on the left correspond to higher similarity bins between translated and original English embeddings (cosine space).

