# OpenReview forum: "Obscuring Data Contamination Through Translation: Evidence from Arabic Corpora"
_ICLR.cc/2026/Conference — Submitted to ICLR 2026_

### Official Review · Reviewer_HxhX · 2025-10-25

**Soundness:** 2
**Presentation:** 3
**Contribution:** 2
**Rating:** 2
**Confidence:** 4

**Summary:**

## Paper Summary

This paper investigates data contamination in multilingual contexts, focusing on how translating benchmarks into Arabic affects contamination detectability and model behavior. The authors fine-tune several open-weight LLMs (LLaMA-3.2-1B-Instruct, Mistral-7B-Instruct-v0.2, Gemma-3-1B-it, and Qwen3-1.7B) on varying proportions of Arabic-translated versions of three benchmarks—MMLU, XQuAD, and MLQA—and evaluate them on the original English benchmarks.

To probe contamination, the authors extend TS-Guessing (a contamination detection method) by introducing choice reordering for multiple-choice questions and token masking for QA tasks. Their experiments show that Arabic translation conceals contamination signals (e.g., surface-form memorization) without truly eliminating their effects. Models still benefit from exposure to contaminated data, especially those with stronger Arabic proficiency.

Based on these findings, the authors propose a Translation-Aware Contamination Detection (TACD) framework that performs contamination analysis across multiple translated benchmark variants. The paper concludes that multilingual contamination remains an overlooked blind spot in current evaluation pipelines and calls for multilingual, contamination-aware benchmarking standards to ensure fairness and reproducibility in LLM evaluation.

**Strengths:**

## Strengths

1. Novel and timely perspective – The paper addresses a crucial and underexplored issue: how translation can mask contamination in multilingual LLM evaluation. This adds depth to ongoing discussions about data leakage, benchmark validity, and fairness in evaluation.

2. Comprehensive background review – The literature section effectively summarizes forms of contamination, existing detection methods, and industrial transparency issues. It situates the work well within prior studies (e.g., Sainz et al. 2023, Li 2023, Golchin & Surdeanu 2023).

3. Clear and well-organized writing – The paper is clearly written, logically structured, and easy to follow. The motivation, methodology, and findings are presented in a coherent narrative, making it accessible even to readers not deeply familiar with contamination detection.

**Weaknesses:**

## Weaknesses

1. Limited quantitative rigor in the TACD proposal – The TACD framework is primarily conceptual. The paper does not empirically demonstrate its effectiveness across multiple languages or benchmark families. Including even a small-scale implementation (e.g., Arabic vs. French) would make the framework more convincing.

2. Insufficient experimental rigor – The main conclusion, that translation masks but does not eliminate contamination, is built on the assumption that TS-Guessing perfectly detects contamination. However, TS-Guessing is itself an imperfect and indirect measure. I suggest the authors evaluate multiple contamination detection methods (such as entropy, dynamic metric) [4,5,6,7] to validate whether the observed masking effect is consistent across different approaches.

3. Incomplete explanation of experimental anomalies – In Table 2, when the contamination percentage is 10%, Mistral-7B-Instruct-v0.2 achieves the best score on XQuAD, and Qwen3-1.7B achieves the best score on MLQA, significantly higher than other contamination levels. It is unclear whether these spikes are due to statistical noise, overfitting to partial contamination, or artifacts from translation alignment. The authors should further analyze or discuss these irregular trends.

4. Limited generalizability of conclusions –
(1) The fine-tuning approach relies on LoRA and PEFT, which may behave differently from full-parameter fine-tuning. It remains unclear whether the same contamination masking effects hold under full fine-tuning.
(2) The evaluation is conducted exclusively on Arabic, a morphologically rich and low-resource language. The findings may not generalize to typologically different languages (e.g., Chinese or Finnish). Expanding to additional languages would strengthen the paper’s external validity.


5. Missing some related work.


[1] Recent advances in large langauge model benchmarks against data contamination: From static to dynamic evaluation (EMNLP 2025)

[2] Leak, Cheat, Repeat: Data Contamination and Evaluation Malpractices in Closed-Source LLMs (EACL 2024)

[3] Data Contamination Calibration for Black-box LLMs (ACL 2024)

[4] ConStat: Performance-Based Contamination Detection in Large Language Models.

[5] DyCodeEval: Dynamic Benchmarking of Reasoning Capabilities in Code Large Language Models Under Data Contamination (ICML 2025)

[6] Towards Data Contamination Detection for Modern Large Language Models: Limitations, Inconsistencies, and Oracle Challenges (ACL 2025)

[7] Detecting Pretraining Data from Large Language Models.

[8] Do Membership Inference Attacks Work on Large Language Models? (COLM 2024)

**Questions:**

1. Do you have any experimental resutls to support the proposed TACD proposal ?

2. If applying other contamination detector, will the conclusion the same?

3. Can you explain the results in Table 2?

4. Does the conclusion generalie to full parameter finetuning and other language?

---

### Official Review · Reviewer_LNUo · 2025-10-29

**Soundness:** 2
**Presentation:** 1
**Contribution:** 1
**Rating:** 0
**Confidence:** 4

**Summary:**

The paper tackles the question of how data contamination can conceal themselves by training with translated data. The authors studied how could Arabic language behave as the "alternative" language to be translated into and from, and how to detect such "translation-based" data contaminations.

**Strengths:**

1. The scope of the problem is important. As more research is done revolving around large language models, contamination could inflate models performance, and lead researchers towards incorrect conclusions when they build their research upon that.

**Weaknesses:**

1. Lack of Novelty. The same idea (translation can cause hard-to-find data contaminations) has been explored in existing research [1] [2]
2. The method author proposed TACD wasn't even evaluated in their experiments (if I read the paper correctly, it's only a proposal)
3. Writing and plotting can use some improvements. You can drop some of that item lists, and images are overly large.

[1]: Yao, Feng, et al. "Data contamination can cross language barriers." arXiv preprint arXiv:2406.13236 (2024).

[2]: Srivastava, Alisha, et al. "OWL: Probing Cross-Lingual Recall of Memorized Texts via World Literature." arXiv preprint arXiv:2505.22945 (2025).

**Questions:**

Please refer to my main concerns.

---

### Official Review · Reviewer_pMR3 · 2025-11-01

**Soundness:** 2
**Presentation:** 3
**Contribution:** 1
**Rating:** 2
**Confidence:** 4

**Summary:**

This paper investigates a form of data contamination where evaluation benchmarks are translated into another language (specifically, Arabic), included in a model's fine-tuning data, and subsequently inflate performance when the model is evaluated on the original English benchmark.

The authors demonstrate that this translation-based contamination successfully "obscures" detection, as standard accuracy metrics misleadingly increase (e.g., in Table 2, MMLU scores for Mistral rise from 0.577 to 0.690).

**Strengths:**

1. Strong, Replicable Methodology: The paper's experimental design is its greatest strength. The controlled study fine-tuning on 0-100% translated data  is clean, and the results are unambiguous.

2. Effective Use of Probing: The adaptation of TS-Guessing with choice-reordering is clever and provides compelling, direct evidence (the high IDR) that memorization is the cause of the performance gains.

3. Valuable Data Point: The specific focus on Arabic provides a useful case study, extending the cross-lingual contamination phenomenon to a non-European, low-resource language.

**Weaknesses:**

1. **Missing Critical Prior Work:** The most significant weakness is the failure compare with existing work [1], which is not just "related"; it established the very phenomenon of cross-lingual contamination that this paper investigates. This omission makes the paper's framing as a novel investigation of a "blind spot"  feel misleading.


2. **Limited Scope:** The study is confined to a single language (Arabic) and three benchmarks. The prior CrossLanguage work was far broader, testing seven languages and demonstrating how the effect varies between language families (e.g., European vs. Asian).


3. **Lack of Comparison:** The prior Cross Language work proposed a new probe called "choice confusion". This paper proposes using an extended "TS-Guessing" probe. A much stronger contribution would have been to compare these two different probes to see which is more effective at detecting this form of contamination.


4. **Vague Framework:** The proposed "Translation-Aware Contamination Detection (TACD)" is acknowledged by the authors to be a "forward-looking blueprint"  and not a concrete, implemented solution. It's an obvious conclusion ("we should check other languages") rather than a technical contribution.

[1] Data Contamination Can Cross Language Barriers, https://aclanthology.org/anthology-files/pdf/emnlp/2024.emnlp-main.990.pdf

**Questions:**

The primary weakness is the missing comparison with [1]. This paper introduced and proved the concept of "cross-lingual contamination". Given this, could the authors please re-frame their contribution, explicitly stating how their work (which uses PEFT and a TS-Guessing probe ) builds upon or differs from this foundational work?

[1] Data Contamination Can Cross Language Barriers, https://aclanthology.org/anthology-files/pdf/emnlp/2024.emnlp-main.990.pdf

---

### Official Review · Reviewer_RQrz · 2025-11-03

**Soundness:** 1
**Presentation:** 1
**Contribution:** 2
**Rating:** 0
**Confidence:** 5

**Summary:**

The authors experiment with finetuning models on english + arabic translations of some benchmarks to understand the impact of translated contamination on top of existing english contamination. They also check if translated contamination can be detected with a previous method(TS-guessing) and show that translated contamination can avoid detection by this method.

While the underlying problem of cross lingual contamination is an interesting one the formulation and experiments in this paper seem to be lacking the quality of design and rigor to answer these questions in a reliable way.

**Strengths:**

The authors select an important problem to work on in a timely manner.

**Weaknesses:**

The experimental design is quite weak from all aspects. Is the objective measuring the presence of cross lingual contamination on top of english contamination, what is the reason behind adding english test example into the fine-tuning mixture? Why not just check the impact on arabic contamination? In the current setting even p=0 is fully contaminated with english test samples.

Why only use TS-Guessing, there are many methods that use model outputs to detect training data membership or contamination. The experimental claim that these models avoid detection seems unrealistic.

Claims around performance and model behaviour in section 4.2 are not backed. Table 2 shows significant changes in performance and while inconsistent it does not show near-flat trends.

The writing is poor, clearly AI generated, tables are not labelled and overall paper structure needs significant improvement.

**Questions:**

No questions since the paper in its current form is quite far from being publishable.

---

### Meta-Review · Area_Chair_oeCc · 2026-01-03

**Summary:**

This paper studies contamination by fine-tuning open-source LLMs on different Arabic datasets and evaluating them on the original English benchmarks. The results show that, while translation can conceal contamination signals, the models can still benefit from contaminated data. The paper further proposed a Translation-Aware Contamination Detection framework.

**Reviewer Concerns:**

The reviewers agree that the problem is interesting and timely. However, there are also many concerns, such as missing critical prior work, lack of comparison, and improving of writing. The authors can carefully read the comments from the reviewers and address them.

**Reviewer Scores:**

The reviewers' scores might not change if they had been able to participate fully in the discussion.

---

### Decision · Program_Chairs · 2026-01-26

Reject